# Correlation between Different Psychological Variables in Women with Fibromyalgia with Symptoms of Neurogenic Inflammation: A Cross-Sectional Study

**DOI:** 10.3390/biomedicines12030671

**Published:** 2024-03-17

**Authors:** Víctor Riquelme-Aguado, Alazne Zabarte-del Campo, Guillermo Baviano-Klett, Josué Fernández-Carnero, Antonio Gil-Crujera, Francisco Gómez-Esquer

**Affiliations:** 1Department of Basic Health Sciences, Rey Juan Carlos University, 28933 Madrid, Spain; guillermo.baviano@urjc.es (G.B.-K.); antonio.gil@urjc.es (A.G.-C.); francisco.gomez.esquer@urjc.es (F.G.-E.); 2Grupo de Investigación Emergente de Bases Anatómicas, Moleculares y del Desarrollo Humano de la Universidad Rey Juan Carlos (GAMDES), 28922 Alcorcón, Spain; 3Department of Basic Medical Sciences, CEU San Pablo University, 28668 Boadilla del Monte, Spain; 4Fisioterapia Oreka CB, 45200 Illescas, Spain; alazne_1990@hotmail.es; 5Escuela Internacional de Doctorado, Department of Basic Health Sciences, Rey Juan Carlos University, 28922 Alcorcón, Spain; 6Cognitive Neuroscience, Pain and Rehabilitation Research Group (NECODOR), Faculty of Health Sciences, Rey Juan Carlos University, 28922 Madrid, Spain; josue.fernandez@urjc.es; 7La Paz Hospital Institute for Health Research, IdiPAZ, 28029 Madrid, Spain; 8Department of Physical Therapy, Occupational Therapy, Rehabilitation and Physical Medicine, Rey Juan Carlos University, 28922 Alcorcón, Spain; 9Grupo de Investigación de Dolor Musculoesqueletico y Control Motor, Universidad Europea de Madrid, 28670 Villaviciosa de Odón, Spain

**Keywords:** fibromyalgia, hyperalgesia, conditioned pain modulation, anxiety, depression, kinesiophobia, catastrophism, chronic pain

## Abstract

Fibromyalgia (FM) is a chronic pain syndrome hypothesized to arise from a state of neurogenic inflammation. Mechanisms responsible for pain, as well as psychological variables, are typically altered in this condition. The main objective of this research was to explore somatosensory and psychological alterations in women with FM. The secondary objective was to carry out a secondary analysis to correlate the different variables studied and delve into the influences between them. The relationship between different psychological variables in fibromyalgia is not clear in the previous scientific literature. Forty-four individuals participated, of which twenty-two were controls and twenty-two were women with fibromyalgia. The main outcome measures were the Numeric Pain Rating Scale, Fibromyalgia Impact Questionnaire, pressure pain threshold, conditioned pain modulation, anxiety and depression symptoms, catastrophizing and kinesiophobia cognitions. The main analysis showed that there is a moderate correlation between the psychological variables of depression and fear of movement and the ability to modulate pain. There is also a moderately inverse correlation between pain catastrophizing cognitions and pain intensity/disability. Multiple moderate and strong correlations were found among the various psychological variables studied. FM patients exhibit somatosensory alterations alongside negative psychological symptoms that influence the experience of pain, and they may perpetuate the state of neurogenic inflammation.

## 1. Introduction

Fibromyalgia syndrome (FM) is a neurogenic inflammation condition characterized by chronic pain as the main symptom, associated with the presence of other symptoms of similar relevance, such as cognitive disorders, fatigue, restless sleep or the presence of psychological symptoms [1,2]. The diagnosis of FM is exclusively clinical since its etiology remains unknown. The clinical status of FM is heterogeneous in the population; even in the same person, the symptoms can fluctuate daily [3]. In Madrid, Spain, 5% of women aged 46 to 60 are affected, according to epidemiological research [4]. The presence of psychological symptoms can aggravate the painful experience suffered by FM patients. Both negative emotional states such as depression and the presence of cognitive distortions such as pain catastrophizing and fear-related movement may be risk factors for suffering from chronic pain for a longer period, increasing the intensity and the impact on functionality [5,6,7,8,9]. Unfortunately, as shown in recent longitudinal epidemiological studies [10,11], psychological symptoms have worsened owing to the health crisis caused by the COVID-19 pandemic, which may have influenced the clinical condition of FM patients [10,11].

Research into the pathophysiological mechanisms responsible for the pain suffered by patients with FM is of great interest in the scientific community [12]. Several alterations in the nociceptive system have been identified, including the transmission, processing and modulation of painful stimuli [13,14,15,16,17,18,19]. One of the most relevant clinical characteristics in patients with FM is the dysfunction in the pain inhibitory systems. It is suggested that the endogenous capacity to modulate pain is reduced, which translates into greater central pain processing [19].

Conditioned pain modulation (CPM) is a dynamic psychophysical test that reflects the capacity of the descending pain modulatory systems to decrease pain [20]. The CPM effect can be quantified by comparing the pain response to a noxious test stimulus applied before and during (or immediately after) a noxious conditioning stimulus in another body region. In healthy individuals, the painful conditioning stimulus should increase the pain threshold by triggering an efficient response from the descending inhibitory system. Nevertheless, results obtained in the CPM show great inter-individual variability, possibly due to differences establishing CPM protocols [21,22]. The CPM test may be reliable to demonstrate a state of neurogenic inflammation present in patients with FM. Thereby, the dysfunction of pain inhibitory mechanisms is evident, which is characteristic of neurogenic inflammation conditions.

On the other hand, emotional psychological symptoms such as anxiety and depression and cognitive psychological symptoms including catastrophism or fear-related movement are shown in neurogenic inflammation conditions [23]. Even though all these are existing variables in patients with FM, their influence on each other remains unclear (e.g., whether or not suffering from anxiety can trigger cognitive alterations such as fear of movement). Furthermore, especially in patients with FM, it is unknown whether these alterations are secondary to a state of chronic pain or whether these alterations might instead cause a condition of neurogenic inflammation and central sensitization. The main objective of this research was to explore somatosensory and psychological alterations in women with FM. The secondary objective was to carry out a secondary analysis to correlate the different variables studied and delve into the influences between them. The role of psychological factors in FM is often studied, although the direction of the relationship remains unclear. Given that the evidence remains weak, increasing knowledge of the relationship between psychological aspects and somatosensory variables of pain can help improve clinical decision-making in patients with FM and help us to find the most appropriate treatment for each case.

## 2. Materials and Methods

### 2.1. Participants

The study involved twenty-two female FM patients (FM) and twenty-two healthy female control subjects recruited through local support group advertisements and presentations. Data collection occurred from February to December 2023 and received approval from the Rey Juan Carlos University Ethical Review Board (2605202012920) following the Declaration of Helsinki. Convenience sampling method was used. All participants provided written informed consent. Inclusion criteria for FM patients included: (1) medical diagnosis of fibromyalgia by a rheumatologist specialist, (2) experiencing pain for over 3 months and (3) fluency in spoken and written Spanish. The criteria of speaking and understanding Spanish correctly did not exclude any participant for any ethnic or social reason. A researcher was present to ensure that participants understood the asked tasks and to answer any questions related to the self-administered questionnaires. Exclusion criteria for FM patients comprised cognitive inability to understand or complete measurement variables. Control participants met criteria for having no pain (NPRS = 0) and fluency in spoken and written Spanish. Exclusions for control subjects included recent musculoskeletal pain episodes within the last 12 weeks and any rheumatologic diseases. A flowchart is shown in Figure 1.

### 2.2. Clinical Status

Pain intensity was assessed using the Numeric Pain Rating Scale (NPRS), a validated tool extensively employed for self-reported evaluation of perceived pain intensity among individuals coping with chronic pain [24]. It consists of an 11-point scale where 0 indicates no pain and 10 indicates the worst pain imaginable. NPRS scores are interpreted as follows: 0 = no pain; 1–3 = mild pain; 4–6 = moderate pain; 7–10 = severe pain. The recent scientific literature strongly recommends the NPRS as the foremost choice, attributing to its heightened sensitivity and consistent measurement of pain intensity [25].

The impact of FM on patients’ daily functioning and the resulting disability was appraised using the Spanish iteration of the Fibromyalgia Impact Questionnaire (FIQ). It is composed of 10 items that allow us to assess the degree of interference from FM symptoms in the person’s daily functioning during the last week. The first four items assess physical function, well-being and work performance using Likert-type scales. The rest of the items are answered using a 10 cm visual analogue scale (VAS) and collect information on pain, fatigue, rest quality, stiffness, anxiety and depression. The score of the Likert scale is transformed to be expressed in a range from 0 to 10. In VAS, the given value is the score for each scale. The global impact index is obtained by adding the transformed scores on the ten scales described, ranging from 0 to 100, 100 being the highest impairment caused by the syndrome. This adapted questionnaire has undergone validation for the Spanish population, demonstrating commendable sensitivity, specificity and internal consistency [26,27].

### 2.3. Pressure Pain Threshold (PPT)

Pressure pain threshold (PPT) is defined as the minimum pressure required, under standardized test conditions, to elicit the slightest sensation of pain. It stands as a reliable gauge for evaluating pain sensitivity. A digital algometer (Model FXD 10, Wagner Instruments, Greenwich, CT, USA) measuring pressure in kg/cm^2^ was employed for this purpose. Pressure was incrementally increased on the right upper trapezius muscle at a rate of one kilogram per second until the subject reported the sensation becoming painful. Each measurement was conducted thrice, and the resulting mean value was used as the final recorded measurement. This methodology has demonstrated notably high intra-examiner reliability (ICC values = 0.97) and substantial inter-examiner reliability (ICC values = 0.79) in the upper trapezius muscle region among healthy individuals. In individuals with FM, this method has been validated to assess mechanical hyperalgesia, yielding an ICC value of 0.88 [28,29,30].

### 2.4. Conditioned Pain Modulation

The cold pressor test paradigm was conducted following these steps: initially, a PPT measurement was taken on the upper area of the right trapezius muscle, as previously described. Subsequently, the left arm was immersed in a constant 12 °C water bath for 60 s. Finally, another PPT measurement was obtained on the upper part of the right trapezius muscle. The Conditioned Pain Modulation (CPM) value is calculated by subtracting the PPT value during the conditioning stimulus from the PPT value without the conditioning stimulus. In healthy individuals whose descending inhibitory pain system works correctly, it is expected that the second algometry measurement shows increased pressure tolerance with respect to the initial measurement before reaching the painful threshold. In chronic pain populations and healthy individuals, the cold pressor test as a conditioning stimulus has demonstrated favorable intra-session reliability (ICC = 0.77 and ICC = 0.64, respectively) [31].

### 2.5. Psychological Variables

Psychological variables were evaluated through self-administered questionnaires, with a researcher present throughout the process to address any inquiries from participants. Various validated scales were employed to assess emotional and cognitive aspects, as detailed below.

#### 2.5.1. Depression

Depressive symptoms were assessed using the Spanish version of the Beck Depression Inventory—Second Edition (BDI-II). Each of the 21 items ranges from 0 to 3, with 63 points being the highest score, where 0 to 13 means minimal depression, 14 to 19 means mild depression, 20 to 28 indicates moderate depression, and 29 points or more indicates severe depression. This validated questionnaire is extensively employed in chronic pain populations and has demonstrated strong reliability (ICC = 0.73–0.86) [32,33].

#### 2.5.2. Anxiety

Anxiety levels were rated using the validated Spanish iteration of the Hospital Anxiety and Depression Scale (HADS), specifically focusing on the anxiety subscale. This subset comprises seven items, each scored from 0 to 3. A cumulative score exceeding 10 points indicates the presence of anxiety, while a score ranging from 8 to 10 denotes a borderline case and a score below 8 indicates an absence of significant anxiety. The reliability of this test was found to be excellent (ICC = 0.85) [34,35].

#### 2.5.3. Pain Catastrophizing

The Pain Catastrophizing Scale (PCS) in its Spanish version was employed to evaluate cognitive distortions related to pain catastrophizing. This is a self-administered questionnaire consisting of 13 items with a score ranging from 0 “Not at all” to 4 “All the time”. It encompasses three dimensions: helplessness (questions from 1 to 5 and 12, regarding the person’s belief in their influence on their pain); magnification (corresponds to questions 6, 7 and 13 and refers to the exaggeration of the threatening properties of the painful stimulus); rumination (corresponds to questions 8 to 11 and refers to the fact that the patient is unable to stop thinking about pain, being unable to get away from the idea). A total score is obtained (ranging from 0 to 52), so a higher score means greater pain catastrophizing. This validated questionnaire holds substantial prominence in the scientific literature and is widely utilized. Notably, the PCS demonstrates excellent reliability specifically among patients with FM, showing an ICC value of 0.94 [36].

#### 2.5.4. Fear-Related Movement

The assessment of cognitive distortions related to fear of movement (kinesiophobia) was conducted using the Spanish version of the Tampa Kinesiophobia Scale (TSK-11). It is an 11-item scale, each with 4 possible answers, where “totally disagree” obtains 1 point and “totally agree” obtains 4 points. Therefore, the total score ranges from 11 to 44 points, a high score being indicative of greater fear of movement/injury, that is, high levels of fear of movement. This scale has been extensively employed among individuals enduring chronic pain, including those with FM, exhibiting commendable psychometric properties with an ICC of 0.85 [37,38].

### 2.6. Statistical Analysis

The data analysis employed the Statistics Package for Social Science (SPSS 25.00, IBM Chicago, IL, USA) with a 95% confidence interval (95% CI), considering *p*-values below 0.05 as statistically significant. To compare differences among nominal variables (e.g., profession or marital status), the Chi-square test was utilized across groups, each consisting of twenty-two participants. Normality tests, specifically the Shapiro–Wilk and Kolmogorov–Smirnov tests, were conducted, revealing no statistically significant differences indicating an abnormal distribution within the data sets. For the comparison of continuous variables between groups, the Student’s *t*-test for independent samples was applied as the statistical test. Subsequently, Cohen’s d was calculated to assess the effect size of the study variables, categorized as small (0.20–0.49), medium (0.50–0.79) or large (>0.8) according to Cohen’s criteria. The correlation between psychological variables (depression, anxiety, catastrophizing and fear-related movement) and psychophysiological measures (CPM, PPT, FIQ and NPRS) was examined using the Pearson correlation coefficient. A correlation coefficient >0.60 indicated a strong correlation; a coefficient between 0.30 and 0.60 indicated a moderate correlation and a coefficient <0.30 indicated a low correlation. A significance level of *p* < 0.05 was set for all statistical tests conducted during the analysis.

## 3. Results

### 3.1. Clinical Status of FM Patients and Pain-Free Controls

Twenty-two participants were pain-free women controls (with a mean age of 48.55 ± 8.19 years), and twenty-two patients were women diagnosed with FM (52.05 ± 8.35 years). According to the FIQ, patients had on average mildly severe symptoms and severe function deficits with 86.49 ± 3.62. The mean pain intensity was 6.05/10 (SD ± 1.88). There were no statistically significant differences between healthy subjects and patients according to the sociodemographic variables shown in Table 1.

### 3.2. Mechanical Hyperalgesia and Conditioned Pain Modulation

Patients with FM presented a reduced PPT, which is indicative of mechanical hyperalgesia (t = 6.5; *p* < 0.001; d = 0.53), as well as showing lower values in the CPM (t = 7.8; *p* < 0.001; d = 0.64), which indicates alterations in the functioning of the descending pain inhibitory system. Data are shown in Table 2.

### 3.3. Anxiety, Depression, Fear-Related Movement and Pain Catastrophism

The independent sample Student’s *t*-test revealed significant inter-group differences. Patients with FM had higher scores on the HADS anxiety subscale (t = 4.3; *p* < 0.001; higher scores on the BDI-II (t = 18.25; *p* < 0.0001, d = 5.8); d = 2.9), higher scores on the PCS (t = 16.1; *p* < 0.01; d = 3.8) and higher scores on the TSK-11(t = 7.4; *p* < 0.01; d = 7.01) compared to the control group. Data are shown in Table 3.

### 3.4. Correlation Analysis

After examining the bivariate relationships between somatosensory and psychological variables, statistically significant correlations were found. Results are shown in Table 4.

## 4. Discussion

The main objective of this research was to explore somatosensory and psychological alterations in women with FM. The secondary objective was to carry out a secondary analysis to correlate the different variables studied and delve into the influences between them.

Women with FM displayed mechanical hyperalgesia and reduced capacity to modulate pain compared to the control group, which is characteristic of populations experiencing chronic pain. Moreover, females with FM display adverse emotional symptoms such as anxiety and depression alongside cognitive distortions related to pain catastrophizing and fear of movement when compared to healthy controls. Depression symptoms exhibited a moderate correlation with pain modulation capacity and a strong correlation with emotional symptoms of anxiety and cognitive distortion related to fear of movement. Anxiety demonstrated a strong correlation with depression, as well as with cognitive distortions regarding pain catastrophizing and fear-related movement. Pain catastrophizing cognitions exhibited a moderate negative correlation with pain intensity (NPRS) and disability (FIQ), whereas they displayed a strong correlation with emotional symptoms of anxiety and cognitive distortions related to fear of movement. Furthermore, fear-related cognitive distortions showed a moderate correlation with altered functioning of pain inhibitory systems, as well as a strong correlation with the presence of negative emotional symptoms of depression and anxiety and pain-related catastrophic cognitions.

### 4.1. Correlations between Somatosensory and Psychological Variables

Patients with FM experience moderate to severe levels of pain intensity and disability. According to Tabach Apraiz et al., the average level of pain intensity is 7.29 [39]. Regarding the validation study of the FIQ, the mean scores may vary depending on the authors (Rivera and González’s study scored 52, whereas Monterde et al. set it at 70.8) [27,40].

In our study, through a secondary correlation analysis, we observed that patients exhibiting higher levels of catastrophizing subsequently reported lower levels of pain intensity and disability. Being that this finding is likely to be surprising, the prior scientific literature warns about potential biases among FM patients in self-administered questionnaires, particularly those undergoing legal processes, as highlighted in the study by Capilla Ramirez et al. [41]. Previous scientific evidence demonstrates that patients with FM in ongoing legal litigation are more consistent in their responses regarding different disability and psychological questionnaires or general health status. Conversely, as shown in the validation study of the FIQ in the Spanish population, those who do not have pending legal litigation do not show such a marked consistency in their responses between questionnaires [40]. Our study shows that catastrophizing pain exaggeration traits does not always correlate with other scales measuring pain intensity or disability in patients with FM. Furthermore, no correlation has been found between high scores on the HADS anxiety questionnaire, higher pain intensity levels (NPRS) and mechanical hyperalgesia (PTT). Previous scientific evidence suggests that patients with anxiety symptoms present greater activation of the medial prefrontal cortex, which is involved in the emotional aspects of pain while they are under a painful stimulus. Nonetheless, our study does not report higher NPRS scores in baseline situations where there was not a painful stimulation, nor when measurements were made with the PPT [42]. These findings suggest that even though patients with FM present high anxiety levels, these symptoms do not influence the detection of mechanical stimuli nor the pain levels reported (NPRS) in rest situations.

In different populations suffering from chronic pain, the CPM test shows reduced activity of descending inhibitory systems [43]. In addition, reduced CPM is also associated with pain in more body regions [44], as could be the case of patients with FM. Owing to the influence of the prefrontal cortex (PFC) and the anterior cingulate cortex (ACC) on the emotional response to pain and pain behaviors, it is hypothesized that psychological factors such as depression or pain catastrophizing can influence the ability to modulate pain [45,46,47]. Conflicting results have been reported regarding the correlation between psychological factors and CPM, both in healthy subjects and in those suffering from chronic pain [48,49]. There is evidence of alterations in the CPM in patients with FM [19], although the scientific literature is scarce when it comes to how psychological variables can influence the ability to modulate pain measured with the CPM paradigm [50]. According to our findings, depression and fear-related movement moderately correlate with reduced functioning of inhibitory systems. In our work, the cold thermal stimulus was used to trigger the conditioning stimulus. It must be considered that there are other methods used to provoke the conditioning stimulus in the CPM test, such as thermal heat stimuli, electrical stimuli or ischemic stimuli. Future studies should evaluate how the psychological variables of patients with FM influence the CPM test performed with different conditioning stimuli.

### 4.2. Correlations between Different Psychological Variables

Previous scientific evidence has found psychological alterations in patients with FM, although the relationship between them lacks certainty. Increasing knowledge of this relationship can help make clinical decisions in each case. Our findings regarding emotional psychological variables (anxiety and depression) align with the prior scientific literature, as demonstrated in the recent study by Henao-Perez et al. [51] One of the remarkable features of our work involves conducting a secondary analysis to delve into the correlation between psychological variables. Our results provide evidence of a strong correlation between anxiety and depression, indicating their frequent co-occurrence in FM patients. Additionally, a moderate correlation was found between heightened anxiety levels and cognitive distortions related to pain catastrophizing or fear of movement. Depressive symptoms exhibited a stronger correlation with fear of movement but not with pain catastrophizing cognitions.

With regard to cognitive psychological variables, our findings also align with the prior scientific literature, revealing cognitive distortions of pain catastrophizing and kinesiophobia in patients with FM. Furthermore, in the secondary correlation analysis, our results agree with those introduced by Koçyigit et al. [9], who found a correlation between kinesiophobia, depression and functional disability (FIQ). In our current study, we observed a correlation between kinesiophobia and other psychological variables (anxiety, depression and catastrophizing), but not with disability. Finally, regarding pain catastrophizing, our results are consistent with the prior scientific literature, indicating elevated levels of pain catastrophizing among women with FM. In this regard, the previous scientific literature has also linked pain catastrophizing with lower levels of physical activity and increased fatigue [52].

### 4.3. Limitations

The primary limitation refers to the absence of blinding among evaluators conducting measurements, resulting in knowledge of participants’ group allocations. Secondly, this study did not assess the medications administered to patients, potentially impacting the outcomes. Lastly, this study encountered a reduced sample size. Future studies that have a larger sample size will study the influences between the variables through linear regression analysis.

### 4.4. Applications for Clinical Practice

The findings of this investigation allow us to better understand the correlation between psychological symptoms and somatosensory variables, as well as the correlations among various psychological factors in FM patients. Thereby, it is suggested that patients with FM who have high depressive symptoms or fear of movement are likely to lack the ability to modulate pain, which must be taken into account when prescribing their treatment. For instance, it is known that physical exercise activates the descending pain inhibitory system. However, when prescribed to patients with FM, their fear of movement can play an important role; being gradually exposed to physical exercise is required to carry out their treatment.

### 4.5. Future Lines of Research

We suggest conducting future longitudinal studies periodically assessing changes in somatosensory and psychological variables in women with FM. This approach would enable a more accurate identification of the interaction between these variables during the course of the disease. Also, future longitudinal studies that subclassify patients with FM and identify the best treatments according to the variables affected in each case are encouraged. Finally, in accordance with the line of research of our research group, it seems appropriate to carry out studies where implicit motor imagery is used in the early stages as a pain treatment technique in patients with FM, since, according to our previous studies, implicit motor imagery is related to the dysfunction of their pain inhibitory systems [53,54]. Consequently, as its implementation does not evoke painful sensations, nor is it influenced by the psychological aspects in patients with FM, we hypothesize that it could enhance their pain coping and subsequently, secondarily influence psychological symptom improvement.

## 5. Conclusions

Depression and fear-related movement are two psychological variables that can influence pain modulation in patients with fibromyalgia. The correlations between the different psychological variables must be taken into account in the clinical setting.

## Figures and Tables

**Figure 1 biomedicines-12-00671-f001:**
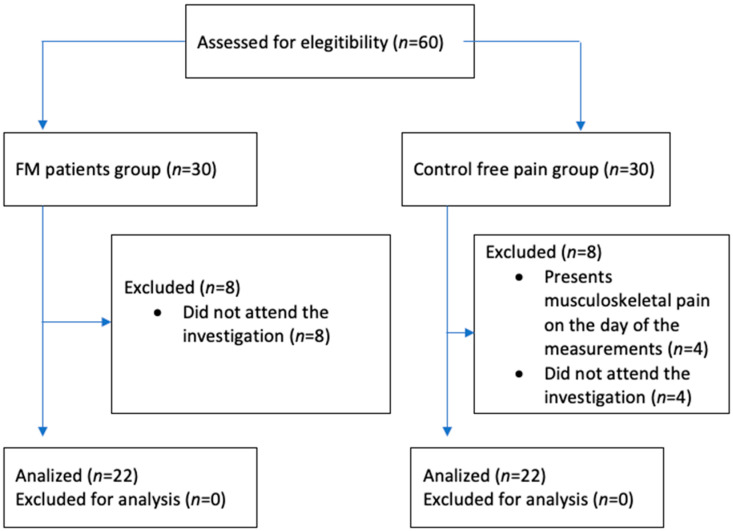
Study design. The subsequent flowchart illustrates the allocation of participants in both the FM study and control groups. Instances that were excluded due to failure to meet selection criteria or withdrawal from the investigation are depicted.

**Table 1 biomedicines-12-00671-t001:** Descriptive statistics in demographic measures (*n* = 44).

Measures	FM (*n* = 22)	Pain-Free Controls (*n* = 22)	*p* Value-Independent Sample Student *t*-Test
Age	52.05 ± 8.35	48.55 ± 8.19	0.84
Pain (NPRS)	6.05 ± 1.88	not measured	not measured
FIQ	86.49 ± 3.62	not measured	not measured

Data are presented as means (SD). Patients in the FM sample present high values of clinical pain intensity on the Numeric Pain Rating Scale (NPRS) and high levels of impact of the disease measured with the Fibromyalgia Impact Questionnaire (FIQ). FIQ and NPRS were administered only to FM participants. FM = Fibromyalgia; NPRS = Numeric Pain Rating Scale; FIQ = Fibromyalgia Impact Questionnaire.

**Table 2 biomedicines-12-00671-t002:** Between-group comparison in psychophysiological measures.

Measures	FM (*n* = 22)	Pain-Free Controls (*n* = 22)	*p* Value Independent Sample Student *t*-Test	T Values
PPT	2.21 ± 0.04	3.21 ± 0.63	0.001 *	6.5
CPM	0.01 ± 0.78	1.5 ± 0.46	0.001 *	7.8

This table shows differences between groups in psychophysiological measures of PPT (pressure pain threshold) and CPM (conditioned pain modulation). The FM group has reduced PPT, indicative of mechanical hyperalgesia and decreased activation of the descending pain inhibitory system. * Indicates values with a *p*-value inferior to 0.05 (statistically significant). FM = Fibromyalgia; PPT = pressure pain threshold; CPM = conditioned pain modulation.

**Table 3 biomedicines-12-00671-t003:** Between-group comparison in anxiety, depression, fear-related movement and pain catastrophizing psychological variables.

Measures	FM (*n* = 22)	Pain-Free Controls (*n* = 22)	*p* Value Independent Samples Student *t*-Test	T Values
Anxiety	11.09 ± 3.02	7.28 ± 2.6	0.001 *	4.3
BDI (II)	37.27 ± 0.7.9	9.86 ± 0.2.1	0.001 *	18.25
Pain catastrophizing	34.11 ± 3.7	15.14 ± 4.02	0.001 *	16.1
Fear-related movement	31.32 ± 3.61	15.27 ± 9.10	0.001 *	7.4

This table show differences between groups in psychological variables of anxiety, pain catastrophism. * Indicates values with a *p*-value inferior to 0.05 (statistically significant). FM = fibromyalgia; BDI II = Beck Depression Inventory.

**Table 4 biomedicines-12-00671-t004:** Correlation coefficients between somatosensory and psychological variables.

Measures	Depression	Anxiety	Pain Catastrophizing	Fear-Related Movement
NPRS	−0.06	0.04	−0.34 *	−0.1
FIQ	0.22	0.14	−0.3 *	−0.02
PPT	−0.02	0.14	0.09	−0.08
CPM	0.36 *	0.26	−0.08	0.30 *
Anxiety	0.64 *	-	0.51 *	0.54 *
Depression	-	0.64 *	0.21	0.61 *
Pain catastrophizing	0.2	0.51 *	-	0.60 *
Fear-related movement	0.61 *	0.54 *	0.60 *	-

This table shows correlation coefficients between somatosensory and psychological variables. * Indicates values with correlation. Coefficient > 0.60 indicated a strong correlation; coefficient between 0.30 and 0.60 indicated a moderate correlation; coefficient < 0.30 indicated a low correlation.

## Data Availability

Data are contained within the article.

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
