# Peer review of "Correlation between Different Psychological Variables in Women with Fibromyalgia with Symptoms of Neurogenic Inflammation: A Cross-Sectional Study"

_biomedicines, 2024, doi:10.3390/biomedicines12030671_

Round 1
Reviewer 1 Report (Previous Reviewer 1)
Comments and Suggestions for Authors
The sample size is too small and only correlation analysis can be performed, resulting in low value of the results.
Comments on the Quality of English Languagegood
Author Response
A word file is attached with the response to the reviewer.

Reviewer 2 Report (Previous Reviewer 2)
Comments and Suggestions for Authors I would like to extend my appreciation to the authors for their thorough and diligent responses to the comments provided during the initial review. The revisions made have significantly strengthened and enhanced the clarity and overall quality of the manuscript. I am pleased to see the authors' careful consideration of the feedback, and it is evident that they have made a concerted effort to align the manuscript with the suggested revisions. I look forward to seeing the final version of the manuscript.Author Response
A word file is attached with the response to the reviewer.

Reviewer 3 Report (New Reviewer)
Comments and Suggestions for Authors
The article by Víctor Riquelme Aguado, titled "Correlation between different psychological variables in 2 women with fibromyalgia with symptoms of neurogenic in-3 flammation: a cross-sectional study” appears to be well structured and therefore I recommend that the paper be accepted in its current for.
Author Response
A word file is attached with the response to the reviewer.

Reviewer 4 Report (New Reviewer)
Comments and Suggestions for Authors
Please refer to the attached file.

Author Response
A word file is attached with the response to the reviewer.

Round 2
Reviewer 1 Report (Previous Reviewer 1)
Comments and Suggestions for Authors
The topic is interesting, and most of the problems have been revised, that is great. However, I still have worriers about the methods, especially for the small sample size.
Comments on the Quality of English Languageit is good
Author Response
see attached word file

Reviewer 4 Report (New Reviewer)
Comments and Suggestions for Authors
Accept in present form
Author Response
See word file

This manuscript is a resubmission of an earlier submission. The following is a list of the peer review reports and author responses from that submission.
Round 1
Reviewer 1 Report
Comments and Suggestions for Authors
The topic is interesting. However, there are some areas in the current version of the manuscript that the authors need to consider or clarify
Background
l I don't see the significance of the research in the background.
Why is this research necessary? The background does not answer this question. Please clarify.
Methods
l The sample size is too small and only correlation analysis can be performed, resulting in low value of the results.
l Is the sampling method convenience sampling or cluster sampling? Please clarify.
l What is the meaning “To compare differences among nominal variables (e.g., profession or marital status), the Chi-square test was utilized across groups, each consisting of seventeen participants.”? How many patients were included in the data analysis? As mentioned earlier, it was 22. Please clarify.
l Th sample size was too small. Does this qualify for statistical analysis? Are the results credible?
l “Subsequently, Cohen’s d was calculated to assess the effect size of the study variables, categorized as small (0.20–0.49), medium (0.50–0.79), or large (>0.8) according to Cohen’s criteria. The correlation between psychological variables (depression, anxiety, catastrophizing, and fear-related movement) and psychophysiological measures (CPM, PPT, FIQ and NPRS) was examined using the Pearson correlation coefficient. Correlation coefficient >0.60 indicated a strong correlation; a coefficient between 0.30 and 0.60 indicated a moderate correlation, and a coefficient <0.30 indicated a low correlation.”
1) Please provide references for classification.
2) From a statistical perspective, the results of Pearson correlation has almost no value in clinical research.
Conclusion
l “Patients with fibromyalgia exhibit somatosensory alterations in pain processing and modulation, alongside negative psychological symptoms that influence the experience of pain and they may perpetuate the state of neurogenic inflammation.”
This conclusion cannot be drawn based on the statistical analysis methods used by the authors.
Comments on the Quality of English LanguageMinor editing of English language required
Reviewer 2 Report
Comments and Suggestions for Authors
I commend the authors on the well-written and organized manuscript that investigates the correlation between psychological variables and somatosensory variables, along with exploring correlations among different variables and their mutual influences. Overall, the study contributes significantly to the understanding of these relationships. I have a few suggestions for minor revisions and seek clarification on a specific aspect of your findings.
Minor Revisions:
§ Lines 212-213: Check for the typographical error
§ Lines 237-238: The authors mentioned that “Emotional symptoms of anxiety demonstrated a moderate correlation with perceived pain intensity on the NPRS”, whereas no statistical significant figure indicative of this is shown in table 4!
§ Consider mentioning the full names of acronyms used in the tables' footnotes. This practice will improve reader comprehension.
§ To provide a more complete understanding of the statistical results, it is advisable to include t values in the tables, in addition to p values. This will contribute to the transparency and interpretability of your findings.
§ Throughout the manuscript, the authors sometimes use the acronym "FMS" to refer to fibromyalgia syndrome or “FM” to refer to fibromyalgia. To ensure consistency, consider consistently using either of the acronyms throughout the entire manuscript. This will enhance clarity and readability.
§ I have several comments and concerns regarding the inclusion criterion of fluency in Spanish for participants.
o I am concerned about the potential introduction of bias by using language inclusion criteria. Please discuss the impact of this criterion on the generalizability of findings and whether it inadvertently excludes specific demographic groups.
o Have you considered alternative approaches to assess language proficiency or ways to design the study that minimize language-related barriers? Please address this to enhance the accessibility of your research.
o The manuscript should explicitly address the ethical considerations associated with language inclusion criteria. Please elaborate on how you have considered the principle of inclusivity and fairness in the recruitment process.
o Clarify how the language criterion is communicated during the informed consent process and within the manuscript. Transparency is essential for reader understanding and participant consent.
o If language proficiency is crucial, provide details on how participants' fluency in Spanish is documented. Additionally, if a specific assessment tool is used, share information about its validity and reliability.
§ In your findings, a negative correlation was observed between catastrophizing and pain intensity and disability. It is intriguing that the authors related this to the inconsistency in responses among patients without legal litigation. It would be beneficial for the readers if you could elaborate on this point. How does the lack of legal litigation influence the relationship between catastrophizing and pain intensity/disability? Providing more context and potential mechanisms for this observation could enhance the interpretation of your results.
§ Consider expanding on the broader context of your findings. Are there specific clinical implications or potential avenues for future research that could stem from these results?
§ Given the complexity of the relationships explored in your study, it might be valuable to suggest potential directions for future research. Are there specific variables or subgroups that warrant further investigation based on your current findings?
Your attention to these suggestions is greatly appreciated and I look forward to the revised manuscript.
Reviewer 3 Report
Comments and Suggestions for Authors
Thank you for entrusting me with the review of this manuscript. I have thoroughly examined your research on fibromyalgia syndrome and its implications for somatosensory and psychological variables.
My feedback is aimed at enhancing the clarity and impact of your work:
1.How was the sample size calculated? If not specified, please mention in the Methods section that it was determined by convenience sampling.
2.Table 1: Ensure that values for the control group are accurately represented, including zeros if applicable. If pain intensity was not measured using the NRS for the control group, this should be clarified. The same applies to the FIQ measurements.
3.Discussion: Avoid repetition with the results section. Condense the presentation of findings by eliminating redundancy and unnecessary elaboration. Focus on providing the most relevant and impactful results without redundant details. Provide a succinct interpretation of the results, highlighting their significance for understanding FMS. Emphasize the key correlations observed between somatosensory and psychological variables and their implications for clinical practice.
4. Very important: Clearly articulate how this research contributes something new compared to previous studies in the field.
5.Conclusion: Expand upon the objectives of the study and clarify the implications of the findings in relation to these objectives. Provide a more comprehensive summary of the key findings and their significance.
Comments on the Quality of English Language
English must be improved by a native speaker to ensure smoother readability. The grammar is generally clear, but there is room for enhancement, as it sometimes resembles Spanish syntax, making it challenging to grasp the intended meaning. Attention should be directed towards improving sentence structure, maintaining consistency in verb tenses, employing parallel structure, using articles appropriately, and ensuring subject-verb agreement. Shortening sentences and maintaining consistent grammar throughout will enhance clarity.
For example, the sentence "In Spain, in the Community of Madrid, the percentage of people affected according to an epidemiological study reaches 5%, being women between 46 and 60 years of age" could be simplified to "In Madrid, Spain, 5% of women aged 46 to 60 are affected, according to epidemiological research."
Reviewer 4 Report
Comments and Suggestions for Authors
Authors presented a cross-sectional study to assess the correlation between different psychological variables inì women with fibromyalgia with symptoms of neurogenic inflammation. This study's primary goal was to show how women with fibromyalgia experience sensorimotor and psychological changes and, after, to correlate the considered variables and investigate the relationships between them. Findings show that negative psychological symptoms impact pain perception of patients with fibromyalgia and may worsen the condition of neurogenic inflammation.
The study met ethical standard and was approved from the institutiona University Ethical Review Board.
The role of psycological variables on FM is often studied (10.1016/j.bbi.2003.11.001; 10.1016/j.ijmedinf.2023.105280) but the direction of the relationship is not clear. Evidence are small and not so robust, authors shoud add this limit to the appropriate paragraph. Moreover, I could suggest to consider this relationship including age and considering the impact onf older female (10.3390/jcm10235552).